# Relative Risk of Bladder and Kidney Cancer in Lynch Syndrome: Systematic Review and Meta-Analysis

**DOI:** 10.3390/cancers15020506

**Published:** 2023-01-13

**Authors:** Anthony-Joe Nassour, Anika Jain, Nicholas Hui, George Siopis, James Symons, Henry Woo

**Affiliations:** 1SAN Prostate Centre of Excellence, Sydney Adventist Hospital, Wahroonga, NSW 2076, Australia; 2Faculty of Medicine and Health, The University of Sydney, Camperdown, NSW 2006, Australia; 3Institute for Physical Activity and Nutrition, Deakin University, Geelong, VIC 3125, Australia; 4College of Health and Medicine, The Australian National University, Canberra, ACT 2601, Australia

**Keywords:** extra-colonic, urology, urinary tract, oncology, tumour, hereditary nonpolyposis colorectal cancer, genetic predisposition, inherited cancer risk, mutated gene, genetics, genome, DNA, malignant, malignancy

## Abstract

**Simple Summary:**

Lynch syndrome (LS) is an autosomal dominant hereditary disorder characterised by germline mutation in one of the four DNA mismatch repair (MMR) genes (MLH1, MSH2, MSH6, PMS2) or deletion of the EPCAM gene. LS predisposes affected individuals to an increased lifetime risk of colorectal (30–73%), endometrial (30–51%), genitourinary cancers (2–20%) and various other malignancies. Although LS-associated upper tract urothelial carcinoma cancer has been extensively investigated, evidence surrounding the role of LS in developing bladder and kidney cancer remains scarce and inconclusive. The aim of this study was to quantify the pooled relative risk of bladder and kidney cancer, summarise the existing evidence on screening and provide a useful recommendation to guide clinicians in identifying and diagnosing at risk individuals.

**Abstract:**

Background: The association between Lynch syndrome (LS) and a higher risk of upper tract urothelial carcinoma is well established, but its effect on the risk of bladder and kidney cancers remains controversial. This review aimed to compare the relative risk (RR) of bladder and kidney cancer in confirmed LS germline mutation carriers compared to the general population. Methods: Medline, Embase, Cochrane Central, and Google Scholar were searched on 14 July 2022 for studies published in English that reported on the rates of urological cancer in adults with confirmed LS germline mutation. The quality of included studies was assessed using Cochrane’s tool to evaluate risk of bias in cohort studies. Random effects meta-analysis estimated the pooled relative risk of bladder and kidney cancer in LS carriers compared to the general population. The quality of the overall evidence was evaluated using GRADE. Results: Of the 1839 records identified, 5 studies involving 7120 participants from 3 continents were included. Overall, LS carriers had a statistically significantly higher RR of developing bladder cancer (RR: 7.48, 95% CI: 3.70, 15.13) and kidney cancer (RR: 3.97, 95% CI: 1.23, 12.81) compared to unaffected participants (*p* < 0.01). The quality of the evidence was assessed as “low” due to the inclusion of cohort studies, the substantial heterogeneity, and moderate-to-high risk of bias. Conclusion: Lynch syndrome is associated with a significant increase in the relative risk of kidney and bladder cancer. Clinicians should adopt a lower threshold for germline mutation genetic testing in individuals who present with bladder cancer. Further studies evaluating the role and cost-effectiveness of novel urine-based laboratory tests are needed. High-quality studies in histologically proven renal cell carcinoma and their underlying germline mutations are necessary to strengthen the association with LS.

## 1. Introduction

Hereditary nonpolyposis colorectal cancer, known as Lynch syndrome (LS), is an autosomal dominant hereditary disorder characterised by germline mutation in one of the four DNA mismatch repair (MMR) genes (MLH1, MSH2, MSH6, PMS2) or deletion of the EPCAM gene (resulting in MSH2 inactivation by promoter hypermethylation) [1]. MMR proteins are vital in correcting base substitution and small insertion–deletion mismatches generated during DNA replication [2]. Failure to repair the mutated simple sequence repeats results in microsatellite instability (MSI) and predisposes individuals to colonic and extra-colonic malignancies [3]. 

LS is characterised by an increased lifetime risk of colorectal (30–73%), endometrial (30–51%), ovarian (4–15%), gastric (up to 18%), small bowel (3–5%), and pancreatic (4%) cancer [1]. On aggregate, urinary tract malignancies represent the third most common LS-associated cancer in men and women (2–20%), for which there are no proven effective screening or prevention strategies [4]. 

Unlike upper tract urothelial carcinoma (UTUC), which has been extensively investigated, included in the Amsterdam II criteria (1999), and widely recognised as part of the LS tumour spectrum [5,6,7], evidence surrounding the role of LS in developing bladder and kidney cancer remains scarce and inconclusive [1]. Joost and Geary et al. reported an increased risk of bladder cancer, particularly amongst MSH2 germline mutation carriers [8,9]. However, Barrow et al. did not observe a significant predisposition to bladder cancer development for any MMR mutation [10]. Skeldon et al. identified synchronous UTUC or UTUC prior to the diagnosis of bladder cancer in approximately 50% of cases, suggesting that bladder cancer may be a continuum of UTUC rather than an isolated event [11]. 

This is the first systematic review and meta-analysis aimed at quantifying the pooled relative risk of bladder and kidney cancer in adults with confirmed Lynch syndrome germline mutations.

## 2. Methods

### 2.1. Search Strategy

The protocol and reporting of this systematic review and meta-analysis are consistent with the 2020 PRISMA guidelines [12]. The systematic review protocol is registered in PROSPERO (CRD42022377098) and can be accessed via the following link (https://www.crd.york.ac.uk/prospero/display_record.php?RecordID=377098) (accessed on 8 December 2022). Medline (OVID interface, 1948 onward), Embase (OVID interface, 1980 onward), Cochrane Central Register of Controlled Trials (Wiley Interface, current issue), and Google Scholar were searched on the 14 July 2022 using broad terms, including combinations of ‘Lynch syndrome’, ‘MMR’, ‘mismatch repair’, ‘hereditary nonpolyposis colorectal cancer’, ‘HNPCC’, ‘UTUC’, ‘upper urinary tract’, ‘urothelial carcinoma’, ‘renal’, ‘bladder’, ‘urological cancer’. A complete search strategy is represented in Appendix A.

### 2.2. Study Selection

Identified records were imported to ENDNOTE X9 citation management software (Clarivate Analytics, Philadelphia, PA, USA) for screening. After removing duplicates, two researchers (AJN and NH) assessed the remaining studies for eligibility. In case of disagreement, reviewers engaged in a discussion before seeking a third opinion from a third reviewer (GS).

The inclusion and exclusion criteria are displayed in Table 1. Briefly, randomised controlled trials (RCTs), cohort studies, and observational studies that reported the relative risk (RR) of at least one type of urological cancer, including kidney or renal, ureter, bladder, penile, and testicular (but not prostate) cancer, in adults with Lynch syndrome (diagnosed via a confirmed germline mutation), and that were published in English, were considered for inclusion.

### 2.3. Data Extraction

We followed the guidelines in the Cochrane Handbook for Systematic Reviews of Interventions [14] to develop a comprehensive data extraction form. The form was used to extract the following data: manuscript title, journal and year of publication, authors’ names and affiliations, funding sources and conflicts of interest, study details (site, country, methodology, recruitment, results, conclusion), and participants’ characteristics. Data are presented in tabular format and discussed narratively.

### 2.4. Quality Assessment of Included Studies

Cochrane’s tool to evaluate risk of bias in cohort studies was used to assess the quality of the studies on the exposed and non-exposed cohorts [15]. The eight domains are summarised in Table 3.

### 2.5. Meta Analysis

Statistical analysis was conducted on R (“metafor” package). Cochran’s Q test was used to assess the assumption of homogeneity of true effect sizes. The degree of variation across studies due to heterogeneity (*I*^2^) was calculated [16], with *I*^2^ ranges between 0% (indicating no heterogeneity) and 100% (indicating high heterogeneity) [17]. If *I*^2^ was ≥50% or *p*-value < 0.10, the random effects model was used to estimate the pooled prevalence of urological cancer in LS carriers compared to the general population. Alternatively, the fixed effects model was used if *I*^2^ < 50% or *p*-value > 0.10. Changes in relative risk (RR) were defined as statistically significant if *p* < 0.05. 

### 2.6. Assessment of Quality of Evidence

The quality of evidence was assessed using Grading of Recommendations, Assessment, Development and Evaluation (GRADE) criteria [18]. Since no RCTs were included in this meta-analysis, all studies were given an initial score of +3 for quality of evidence. Risk of bias was assessed in accordance with the recommendations of the Cochrane Collaboration [14] as explained above. Serious risk of bias, inconsistency (e.g., due to heterogeneity (*I*^2^ > 50%), and imprecision (e.g., when the 95% CI of the overall effect overlaps the line of no effect) incur the loss of a point [19,20]. The final GRADE score consisted of the sum of all categories [21].

## 3. Results

### 3.1. Study Selection

The search yielded 1839 records (Figure 1). Duplicates (1099) were removed. Five hundred and forty-nine (549) records were excluded by screening titles and abstracts. One hundred and ninety-one (191) full texts were assessed for eligibility. One hundred and eighty-six full-text articles were excluded (reasons shown in Appendix A). Five studies were included in this review. Study characteristics are summarised in Table 2.

### 3.2. Study Characteristics

Table 2 lists the detailed characteristics of the five included studies. A total of 7120 confirmed LS carriers from 6 countries were included. Two studies took place in Europe (Finland, Germany, and the Netherlands), two in the USA, and one was an international study that was carried out in three countries (Australia (primarily), Canada, and USA). Two studies reported the relative risk of both bladder and kidney cancer, two studies reported on bladder cancer, and one study reported the relative risk of kidney cancer. The majority of studies were recent, with one study being published within the last 5 years, three studies being published within the last decade, and one study being published more than 10 years ago. 

## 4. Risk of Bias

Overall, the methodological quality was fairly consistent among all five studies (Table 3). The exposed and non-exposed groups from four studies were drawn from the same population. However, we had to downgrade Win et al. (2012) in this domain (D1) as the authors were unable to determine which population subgroups were included under the umbrella term “Australasian” [25]. We can be fairly confident in regard to the assessment of exposure (D2) for all five studies, given that the data were extracted from major registries. Although we cannot be entirely certain about the absence of a urinary malignancy in the exposed population at the start of the study (D3), we can be fairly confident that this did not exist at the start of the study, given that LS carriers are 10× more likely to present with a colorectal tumour than a genitourinary tumour [26,27]. The greatest methodological limitation in all five studies was that they did not match all variables associated with the outcome, and were thus assessed accordingly for the pertinent domain (D4). This is particularly important when assessing the relative risk of bladder cancer, for which several environmental risk factors exist [4]. Overall, we can be confident in the presence of the most important risk factors (age, sex, and MMR mutation (particularly EPCAM/MSH2)) in all the studies included (D5). We can be fairly confident with the assessment of outcome for the increased RR of bladder cancer in LS carriers compared to the general population, but remain reserved from drawing strong conclusions of association for kidney cancer due to the ambiguity of how “renal/kidney mass” was defined (D6). Of the three studies included in the kidney cancer meta-analysis, only Aarnio et al. (1999) had explicitly distinguished renal cell adenocarcinoma (RCC) from renal pelvis urothelial cancer [22]. All five studies had an adequate follow-up period of over 5 years (D7). Domain 8 could not be ascertained from any of the studies, as these were all observational and not interventional studies.

## 5. Evidence Synthesis

### 5.1. Bladder Cancer

Four studies (*n* = 6760) were included in the meta-analysis assessing the RR of bladder cancer in LS germline mutation carriers compared to the general population [4,23,24,25]. Overall, LS carriers displayed 7.5 times greater RR of developing bladder cancer compared to unaffected participants (RR: 7.48, 95% CI: 3.70, 15.13), which was statistically significant (*p* < 0.01) (Figure 2). Of note, there was substantial heterogeneity between studies (*I*^2^ = 83%), and all of the four included studies were considered to have moderate risk of bias. 

### 5.2. Kidney Cancer

Three studies (*n* = 4634) were included in the meta-analysis assessing the RR of kidney cancer in LS germline mutation carriers compared to the general population [4,22,25]. Overall, LS carriers displayed a nearly four times greater RR of developing kidney cancer, compared to unaffected participants (RR: 3.97, 95% CI: 1.23, 12.81), which was statistically significant (*p* = 0.01) (Figure 2). Heterogeneity among all three studies was considered as moderate (*I*^2^ = 77%), with two of the three studies deemed to have a high risk of bias, given the ambiguity concerning the histological origin of “renal/kidney mass” [4,25].

Overall, the initial evidence was assessed as “low” due to the observational nature of the studies included. A further point was deducted due to the inconsistency arising because of the “substantial” heterogeneity. Therefore, the overall score for the level of evidence was “very low”.

A forest plot was generated depicting the RR of bladder cancer (top) and kidney cancer (bottom) in Lynch syndrome germline mutation carriers compared to the general population. The size of the grey squares is proportional to the sample size of each included study; studies with 95% CI (horizontal line) crossing zero (vertical line of no effect) are inconclusive; powerful studies (i.e., studies with more participants) display narrower 95% CI; the diamond indicates the summary relative risk in the combined sample; the width of the diamond indicates its 95% CI. A statistically greater relative risk in adults with LS germline mutation is apparent, which is 7.5 times greater than the general population in regard to bladder cancer, and 4 times greater regarding kidney cancer. The data present substantial heterogeneity. 

95% CI = 95% confidence interval; df = degrees of freedom; RR = relative risk; SE = standard error.

## 6. Discussion

This is the first systematic review and meta-analysis to report on the RR of bladder and kidney cancer in adults with confirmed LS germline mutations. Our meta-analyses on the RR of bladder (*n* = 6760) and kidney cancer (*n* = 4634) demonstrates a statistically significant increased risk of developing bladder and kidney cancer (7.5× and 4×, respectively) in LS carriers compared to the general population. The results, albeit statistically significant and of a great magnitude in terms of the sample size of the pooled participants, need to be tempered due to the high heterogeneity of the included studies and the moderate-to-high risk of bias of the included studies.

Overall, our results agree with major European and American recommendations that bladder cancer should be regarded as major extra-colonic malignant manifestation in LS carriers, and supports improved screening strategies [7,28]. Although our results also suggest a higher relative risk of kidney cancer in LS carriers, this needs to be interpreted with caution given the uncertainty regarding the histological origin of the reported kidney/renal masses in the included studies.

### 6.1. Bladder Cancer

We suspect that the controversy surrounding the association between bladder cancer and Lynch syndrome from previously reported retrospective studies stems from population selection bias. These studies recruited participants based on low-sensitivity screening tools (Amsterdam Criteria and Bethesda Guidelines) rather than proven genetic mutation carriers, resulting in an overrepresentation of unaffected individuals and, ultimately, equivocal results. Our study has the advantage of only including proven LS mutation carriers and the first meta-analysis demonstrating an increased relative risk of developing LS-associated bladder cancer compared to sporadic disease in the normal population. The mean age for bladder cancer in our study was between 57 to 64 years. Numerous early studies have established a disproportionately increased risk of developing bladder and upper tract urothelial cancer in LS carriers with pathogenic germline variants in EPCAM/MSH2 [11,24,29]. This is echoed in more recent data from prospective LS databases demonstrating a cumulative incidence of bladder cancer (at age 75) of 8.1% among EPCAM/MSH2 carriers compared to 4.1% in MLH1 carriers [24,30]. Another large cross-sectional study of more than 3800 confirmed LS carriers found that personal history of urinary tract cancer amongst LS carriers was independently associated with EPCAM/MSH2 germline variants, but not MLH1, compared to LS carriers without a personal history of urinary tract cancer [4]. We were unable to ascertain tumour stage and grade of bladder cancer from the studies included in our meta-analysis. However, a study of 321 LS carriers with a median follow-up of 9 years found that LS-associated bladder cancer was predominantly non-invasive (90%) and high-grade (71%) [11].

### 6.2. Kidney Cancer

Although our findings support previous registry studies describing an increased risk of kidney cancer in LS carriers, we remain reserved from drawing strong conclusions due to the ambiguity in the histological origin of these defined kidney/renal masses [30,31]. Although renal cell carcinoma is intrinsically heterogenous in its histological variants, distinction relating to the overarching term kidney cancer must be emphasised. Only one study by Aarnio et al. (1999) included in our meta-analysis defined kidney cancer as renal-cell adenocarcinoma (particularly amongst MLH1 mutation carriers), whereas the remaining two studies were unspecified kidney/renal masses. Interestingly, a subsequent pathology study by Ararnio et al. (2012) found that renal cancer amongst MLH1 mutation carriers was low (1.3%), and that MSI was infrequent despite deficient MMR upon immunohistochemistry analysis, suggesting that renal cancer may not be related to the LS tumour spectrum [32]. A recent paired germline and MSI testing study on 15,000 cancer patients demonstrated that only 11/458 (2.4%) of biopsy-proven RCCs demonstrated MSI, none of which had an identified LS germline variant. The two RCCs with the MSH6 germline lacked MSI, suggesting differing aetiology. In contrast, the same study found MSI in 5.8% of bladder and UTUC patients, of whom 12/32 (37.5%) were confirmed LS carriers [33]. In another study of 21 patients comparing hereditary UTUC to sporadic UTUC, four upper tract masses were reclassified as RCC based on final histopathology. Hypermutation or loss of heterozygosity was not detected in the LS germline mutation in these four RCC samples that were collected from individuals with LS, suggesting that pathogenesis of these tumours was not related to LS diagnosis [34]. We believe that further studies on histologically proven RCCs, and their underling germline mutations, are necessary to solidify a potential association.

### 6.3. Upper Tract Urothelial Carcinoma

Upper tract urothelial cancer represents the third most common malignant manifestation in LS carriers [1]. Similar to bladder cancer, the cumulative lifetime risk, at 80 years of age, of developing hereditary UTUC is mutation-dependent, and strongly in favour of EPCAM/MSH2 germline variants (2.2–28%) compared to MSH6 and MLH-1 (0.2–5%) [29,35]. The peak age of presentation ranges between 50–64 [29]. A recent comprehensive review from the European Association of Urology reported a 14-fold increase in LS-associated UTUC compared to the general population, which further increased to 75-fold among MSH2 mutation carriers [28]. Despite the overwhelming clinical and pathological evidence supporting UTUC as part of the LS spectrum of extra-colonic malignancy [6,8,11,24,26,29,32,34,35,36,37,38,39,40,41,42], the data were too heterogenous to conduct a meta-analysis, and only two studies compared hereditary to sporadic UTUC. Similar to our findings, MSH2 germline mutation occurred in 91% and 100% of the patients with Lynch syndrome [34,40]. Unlike bladder cancer, LS-associated UTUC had an equal sex distribution compared to the 2:1 (male:female) preponderance in the general population, occurred a decade earlier (*p* < 0.0001 and *p* = 0.005), and affected individuals with no smoking habit (*p* = 0.03), compared to sporadic UTUC [34,40]. Ureteral involvement was observed in approximately 50% of LS-associated UTUC cases compared to the renal pelvis (65%) in the general population, representing a statistically significant difference in location of the sentinel lesion (*p* = 0.001 and *p* = 0.001) [34,40]. No differences were observed between LS-related and sporadic UTUCs according to tumour size, grade, or stage [34,40]. 

### 6.4. Screening Tools in Clinical Practice for Patients with Lynch Syndrome

Patients with sporadic urinary tract cancers can harbour Lynch syndrome. In a study of 115 patients with sporadic upper tract urothelial cancer, 5.2% were subsequently diagnosed with Lynch syndrome germline mutations [26]. The traditional Amsterdam II Criteria and revised Bethesda Guidelines lack the sensitivity (22% vs. 82%) and specificity (98% vs. 77%) to be considered adequate screening tools [43,44,45]. 

Detection of MSI by polymerase chain reaction (PCR) or evidence of deficient MMR protein (dMMR) by way of immunohistochemistry (IHC) are the molecular screening tools used to aid in the diagnosis of LS [46,47]. Although MSI-PCR is the earliest LS screening test, IHC is more cost-effective, readily available, and with comparable sensitivity and specificity (63% and 88%) in LS-associated UTUC [37,38,48]. A recently published systematic review and meta-analysis concluded that IHC screening for dMMR in unselected UTUC cohorts resulted in a diagnostic yield of 4.7% (51/1087) for Lynch syndrome [49]. This diagnostic yield is similar to that observed in universal IHC screening programs for CRC in LS, which have demonstrated significant survival benefits [50,51]. The extrapolated cost for IHC screening programs for UTUC based on incidence would be approximately 1/10 of the current CRC screening programs (2:100,000 persons vs. 29:100,000 persons) [37]. Data scarcity limited their ability to evaluate MSI-PCR as a screening tool and reported significant variability in concordance with IHC, which is likely due to intrinsic disease factors, small sample size, and poor study design [49]. In the current environment, IHC may have better real-world application as a universal screening test, but its accuracy is affected by the staining process, quality of staining antibodies, and user interpretation [52,53]. Most importantly, dMMR is only a proxy of MSI status, and approximately 10% of high-frequency MSI (microsatellite instability in >30%) displays intact MMR staining and localization [54,55].

MSI status by PCR remains the standard for determining MSI frequency and phenotype [56]. High-frequency MSI (H-MSI) supports LS suspicion (regarded as microsatellite instability ≥ 30% or in at least ≥2 of 5 or ≥4 of 15 DNA markers) [54]. While dMMR is strongly associated with LS-associated colorectal (98%) and endometrial (94%) cancer, concordance is much lower in LS-associated urothelial cancers (23%) [8,57,58,59]. The EAU Young Academic Urologists and the Global Society of Rare Genitourinary Tumours (GSRGT) recommend MSI-PCR as the initial molecular screening test for patients suspected of having LS-associated UTUC [28], reserving IHC for those with H-MSI to help identify the localization of gene involvement and distinguish sporadic mechanisms of H-MSI observed in up to 15% of sporadic cancers. This recommendation is evidence-based but derived from limited, older data [56,60,61].

Interest in MSI detection has surged given its diagnostic, prognostic, and therapeutic clinical potential in the era of immunotherapy [55]. H-MSI is predictive of the efficacy of immune checkpoint programmed cell death-1 protein (PD-1) antibodies pembroluzimab, nivolumamb, and, more recently, combination therapy with nivolumab and ipilimumab (CTLA4 inhibitor) [55]. A systematic review by Therkildsen et al. on immune checkpoint therapy for the treatment of LS concluded that response rates are comparable to those of sporadic MSI cancer patients [57].

Conventional molecular testing failure has been reported to occur in approximately 10% of LS-related colorectal cancer patients [62,63]. In the setting of a negative or normal molecular test but strong clinical suspicion of LS, referral for genetic testing is recommended [7,64]. Cost-effectiveness for various LS diagnostic strategies has only been reported in colorectal cancer [63,65], and no current data on cost-effectiveness amongst LS-associated UTUC exists [28]. Yu et al. provides a comprehensive summary on the developments aimed at improving the sensitivity of MSI detection in tissue and liquid biopsies [55]. 

Multiplex PCR of markers and capillary electrophoresis (PCR-CE) is now regarded as the gold standard for MSI detection in colorectal cancer [66]. Using a pentaplex mononucleotide panel coupled with CE, markers are amplified using fluorescent primers. This enables the inspection of simultaneous targets with a resolution of up to a single base difference [66,67]. A drawback to this technique is that during the PCR process, slipped strand mispairing (‘stutter’) may occur, resulting in false negatives and a limit of detection (LOD) of 10% tumour content [67]. Protocol modifications in the form of nuclease-assisted minor allele enrichment with probe overlap (NaME-PrO) and nuclease-assisted microsatellite instability enrichment (NaMSIE) have reduced LOD to 0.5% [68,69]. These methods can be applied to tissue and liquid biopsies as well as fixed formalin samples [68].

Droplet digital PCR (ddPCR) for MSI detection is an alternative quantitative approach that has been demonstrated to quantify MSI to 0.1% mutant frequency [70]. The technique utilises two fluorescent hydrolysis probes to segregate wild type from mutant DNA, thus enabling the quantification of the latter. Concordance of this technique with gold-standard PCR-CE was observed in 100% of colorectal cancer and 93% of other cancers (none were genitourinary) [70]. Furthermore, ddPCR can be used in tissue and liquid biopsies as well as fixed formalin samples [71]. 

Massively parallel next-generation sequencing (NGS) is rapidly altering the landscape for MSI detection. NGS enables simultaneous MSI analysis of thousands of loci and can quantify MSI to a sensitivity of 0.05%, which is compatible with MSI detection for cell-free DNA (cfDNA) testing [72,73,74,75]. The same NGS assay also provides information on somatic gene profiling and tumour mutational burden to improve the predictive selection of patients for immune checkpoint inhibitors [76,77]. Unlike PCR-CE, NGS is not reliant on preselected microsatellite markers, and may have better practical applicability to a wider array of cancers. Several bioinformatic algorithms using different MSI detection strategies have been reported and succinctly tabulated (Table 1) by Yu et al. [55]. 

MSI sensor is one such software that has been utilised for investigating the genomic differences between within UTUC and bladder cancer [78]. Audenet et al. reported a sensitivity and specificity of 92% and 83% vs. 83% and 97%, respectively, for an MSIsensor score of ≥3 vs. ≥10, while investigating the genomic differences between UTUC and bladder cancer in a 195-patient cohort using NGS technology. Although no MSIsensor score threshold consensus exists, some authors define high MSI as an MSIsensor score of ≥3.5, while others consider a score of ≥10 to be consistent with LS [79,80]. Minimally invasive MSI detection from blood sampling remains technically challenging due to the dilutional effect of normal circulating DNA. However, several MSI detection algorithms in combination with exome sequencing or targeted resequencing panels have enabled MSI detection in cfDNA. One such approach is the NGS-based Inter-Alu-PCR, which successfully distinguished MSI-H from MSS or healthy donors using 0.1 ng cfDNA [73]. Although promising and potentially scalable, further research in this field is needed. We were unable to identify any studies evaluating novel non-invasive urine tests for LS-associated bladder cancer such as Cxbladder and Nuclear Matrix Protein 22 (NMP22). 

### 6.5. Screening Protocols in Clinical Practice for LS-Associated UTUC

There is an urgent need to recognise and develop adequate screening protocols given the increased risk of urothelial cancer in LS carriers. To date, no consensus on screening methodology and starting age exists. Six studies have previously reported on the screening recommendations for individuals with LS, which have been succinctly summarised in a recent comprehensive review and consensus paper by Lonati et al. (Figure 3) [28]. Screening starting age ranged between 25 and 50 years [35,81]. Annual or bi-annual urinalysis and urine cytology was recommended in four of six screening studies [81,82,83,84]. However, long-term surveillance of a 977-patient cohort by Myrhøj et al. concluded that using urine cytology should not represent a proper screening method for LS carriers due to poor sensitivity (29%) and high false-positive (10x greater) [85]. Among patients who had microscopic haematuria, 41% had no underlying urological cancers [86]. A systematic review and expert panel recommendation by an American research group [7] suggested that urologists should suspect UTUC and perform clinical, tissue, and genetic testing for LS if patients present with any UTUC:Before age 60 years;Possess a family history of UTUC, CRC, or endometrial cancer before age 60 years;Possess a personal history of colorectal or endometrial cancer.

Mork et al. were unable to justify upper tract CT imaging with contrast for all patients considering the cost and radiation risk for such a low diagnostic yield [7]. However, in high-risk individuals, such as MSH2 carriers or those with a family history of previous LS malignancy, a reasonable modification would be to perform annual or bi-annual CT urograms (CTU) [7]. A magnetic resonance urogram and renal ultrasound combined with retrograde studies present feasible alternatives; however, ultrasound alone has low sensitivity [7]. The European Association of Urology guidelines on UTUC 2021 have finally provided a flow chart (Figure 4) and ratified molecular testing among UTUC and any of the following [28]:Under 65 years of age;UTUC individuals with a personal history of cases on the LS tumour spectrum;UTUC individuals with one FDR, under 50 years of age, with LS-related cancer;UTUC individuals with two FDRs with LS-related cancer, regardless of age.

According to these EAU guidelines, a patient-based checklist (Figure 5) has been developed to be used by urologists for all UTUC cases to aid in identifying those deserving further investigations [28]. The EAU Young Academic Urologists and the Global Society of Rare Genitourinary Tumours (GSRGT) now recommend a screening protocol for all LS patients starting from 45–50 years or 5 years prior to the earliest age of diagnosis in patients with a family history of UC before the age of 45 years [28]. This includes annual urinalysis and urinary cytology and biennial abdominal ultrasound. MSH2 carriers or with family history of urothelial carcinoma should be alternating ultrasound and abdominal CT scans annually [28]. 

**Figure 3 cancers-15-00506-f003:**
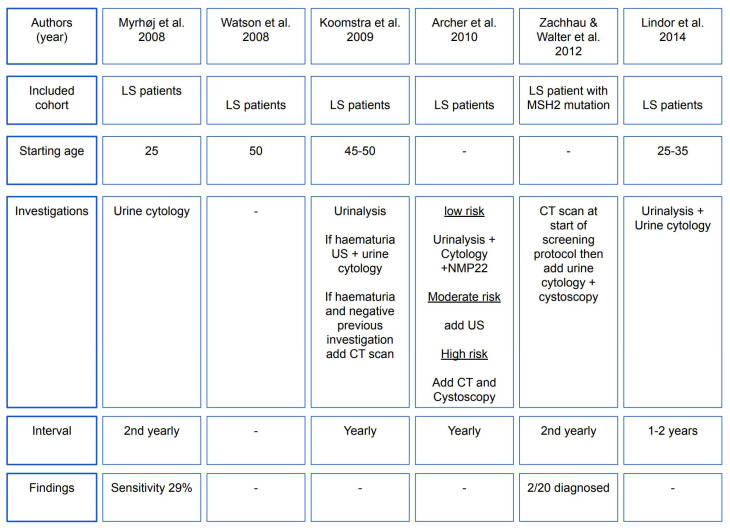
Flow chart summarising screening strategies for the detection of upper tract urothelial carcinoma among individuals with Lynch syndrome [35,81,82,83,84,85]. Adapted from the European Association of Urology—Young Academic Urologists and Global Society of Rare Genitourinary Tumours [28].

**Figure 4 cancers-15-00506-f004:**
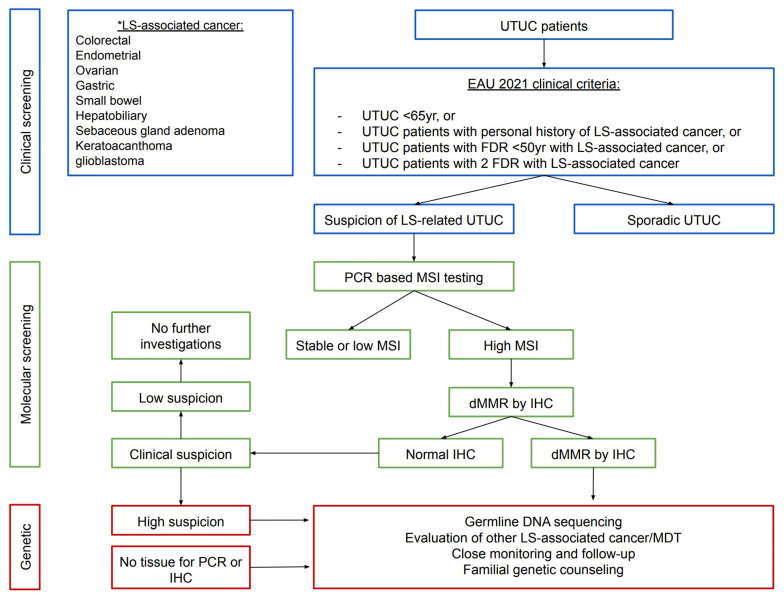
Flow chart for diagnosis of Lynch syndrome in individuals with upper tract urothelial carcinoma. Adapted from the European Association of Urology—Young Academic Urologists and Global Society of Rare Genitourinary Tumours [28]. * LS-associated cancer.

**Figure 5 cancers-15-00506-f005:**
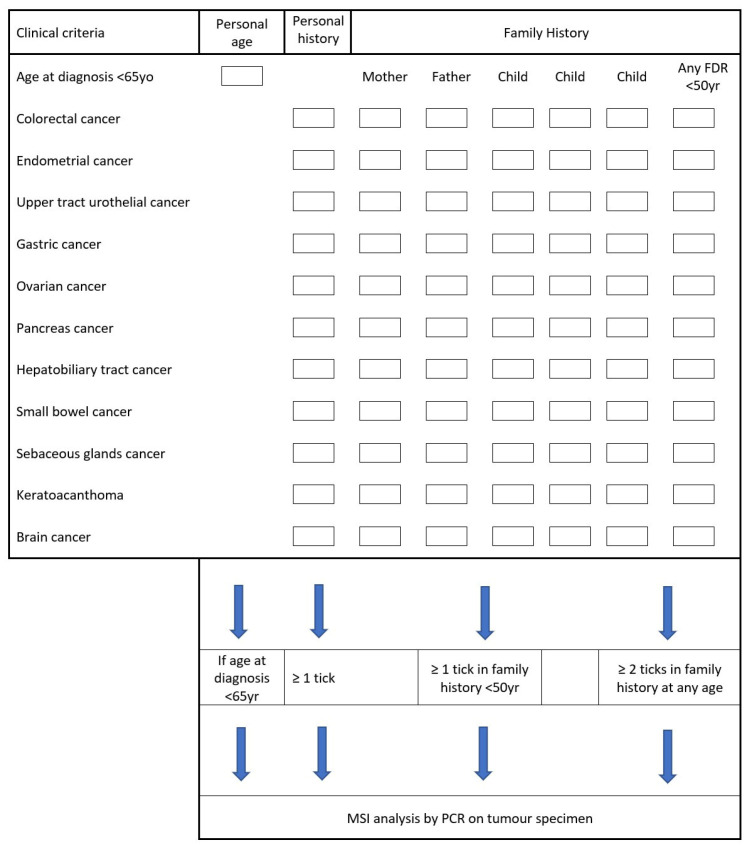
Patient-based checklist to assist the urologist in identifying individuals with upper tract urothelial carcinoma requiring molecular testing for Lynch syndrome. Adapted from the European Association of Urology—Young Academic Urologists and Global Society of Rare Genitourinary Tumours [28].

### 6.6. Limitations

The main strength of this systematic review and meta-analysis is the adoption of stringent methodology outlined in the PRISMA and Cochrane guidelines. This review also has limitations. First, searches were restricted to articles published in English. However, reports indicate that the difference is only 1 out of every 36 meta-analyses when the non-English publications are included [87]. Second, there was great heterogeneity, but this is expected when synthesising data from different countries and healthcare systems. Third, there was a moderate-to-high risk of bias of included studies, and the evidence was assessed as “very low” overall due to the heterogeneity and risk of bias; this is somewhat expected considering that no relevant randomised controlled trials were identified in the literature, and we had to rely on observational studies. Fourth, we were unable to obtain information about grade and stage of disease from the included studies, and therefore, were unable to conduct subgroup analyses by grade and stage of disease that could have provided further insight. Fifth, despite numerous studies reporting on LS-associated UTUC, we were only able to perform a meta-analysis to determine the relative risk of kidney and bladder cancer. Powering studies investigating LS-associated UTUC remains a major hurdle given that newly diagnosed cases are approximately 200–400 per year, or roughly 1/10 of the number of LS-associated colorectal cancer cases [26,27]. Global effort to create pooled registry databases is needed to overcome this. As previously mentioned, despite an extensive search, only a small number of studies were eligible for analysis, and the majority were retrospective. 

We acknowledge that epigenetic and microRNA regulation are important and potentially independent pathways for LS-associated tumour pathogenesis [88,89,90]. Unfortunately, these studies did not meet the selection criteria, and were not evaluated in this study. However, this remains an evolving area of great interest that warrants further investigation.

Lastly, uncertainty in the histological origin of the reported kidney/renal masses remains a major shortcoming of this study. The distinction between histologically proven renal cell carcinoma and the umbrella term kidney cancer was not well represented in any of the included studies, thus limiting our ability to draw confident inferences about the association of RCC in LS, and potentially overestimating our result. Renal cell carcinoma is intrinsically heterogenous in its histological variants. However, it appears that major registries may have not placed emphasis on adequately separating RCC from urothelial carcinoma of the kidney, resulting in contaminated datasets.

## 7. Implications for Practice and Future Research

To date, existing screening tools for bladder cancer and UTUC remain poor, and stringent cost analyses of novel non-invasive urine-based laboratory screening tests on this population group are needed prior to widespread uptake and incorporation into screening recommendations. Therefore, clinicians should adopt a lower threshold for germline mutation genetic testing in people who present with bladder cancer (i) under the age of 65 years, (ii) with a personal or family history of LS-associated tumours, and (iii) without other environmental/occupational risk factors. Further studies on histologically proven RCCs, and their underlying germline mutations, are necessary to strengthen the association with LS. 

## 8. Conclusions

This study addresses a longstanding gap in the literature surrounding the role of Lynch syndrome in the development of bladder and kidney cancer. Lynch syndrome is associated with a significant increase in the relative risk of kidney and bladder cancer. Clinicians should adopt a lower threshold for germline mutation genetic testing in individuals who present with bladder cancer. Further studies evaluating the role and cost-effectiveness of novel urine-based laboratory tests in screening pathways are needed. High-quality studies on histologically proven renal cell carcinomas and their underlying germline mutations are necessary to strengthen the association with LS.

## Figures and Tables

**Figure 1 cancers-15-00506-f001:**
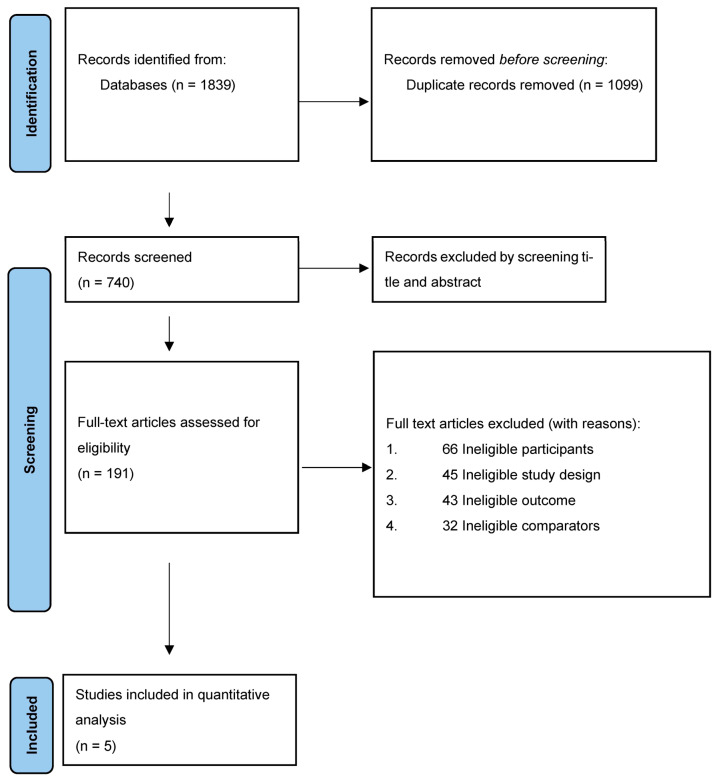
Preferred reporting items for systematic reviews and meta-analyses (PRISMA) flow chart. Five studies eligible for meta-analysis.

**Figure 2 cancers-15-00506-f002:**
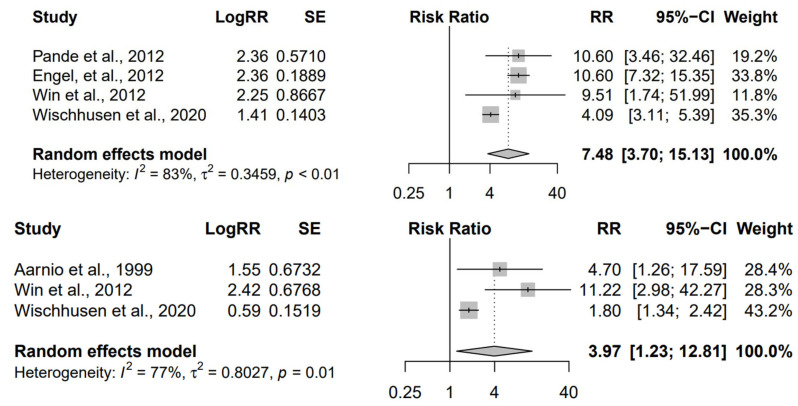
Meta-analysis [4,22,23,24,25].

**Table 1 cancers-15-00506-t001:** Study eligibility criteria.

Inclusion Criteria
Studies reporting the relative risk (RR) of at least one type of urological cancer, including kidney or renal, ureter, bladder, penile, and testicular, but not prostate cancer
Adult participants (≥18 years) with Lynch syndrome
Lynch syndrome diagnosed via a confirmed germline mutation in one or more of MLH1, MSH2, MSH6, EPCAM, or PMS2 upon genetic testing
Studies reporting the rates of urological cancer per participant, since an individual might have more than one cancer
Exclusion criteria
Studies that assessed Lynch syndrome using inconclusive laboratory tests, such as immunohistochemistry, or clinical criteria, such as the Amsterdam criteria
Non-clinical studies e.g., pathology studies
Non-English manuscripts
Commentary or editorial manuscripts
Studies only reporting the rates of prostate cancer that have been discussed by a previous systematic review [13]

**Table 2 cancers-15-00506-t002:** Study characteristics.

Author (Year)	Study Design	Country	Registry/Database	Sample Size (Confirmed MMR Carriers Only)	Male	Mean Age at Diagnosis (Any Cancer)	Ethnicity	Results (SIR (95% CI), *p*-Value)
Aarnio et al. (1999) [22]	Retrospective—Cohort	Finland	Finnish cancer registry	360	Not reported	Not reported	Caucasian (100%)	Increased risk of RCC cancer (4.7 (1.0, 14.0), *p* < 0.005)
Pande et al. (2012) [23]	Retrospective—Cohort	USA	MD Anderson cancer centre database	368	151	41.2 (SD 10.6)	Caucasian 84%	Increased risk of bladder cancer (10.6 (2.9, 27.2), *p* < 0.05)
							African American 7%	
							Hispanic 7%	
							Other 2%	
Engel et al. (2012) [24]	Retrospective—Cohort	Germany and the Netherlands	German HNPCC consortium and the Netherlands foundation for the detection of hereditary tumours	2118	1011	M:53 (IQR 25–73) F:55 (IQR 23–71)	Caucasian 100%	Increased risk of bladder cancer in men (8.5 (5, 13.5), *p* < 0.005) and women (16.2 (8.6, 27.8), *p* < 0.005)
Win et al. (2012) [25]	Prospective—Cohort	Australasia (primary)/USA and Canada secondary	The Colon Cancer Family Registry	446	205	62 (IQR 55–68)	Caucasian 95%	Increased risk of kidney cancer (11.22 (2.31, 32.79), *p* < 0.001) Increased risk of bladder cancer (9.51 (1.15, 34.37), *p* < 0.09)
							Asian 2%	
							Hispanic 1%	
							Middle east 0.5%	
							Other 1.5%	
Wischhusen et al. (2020) [4]	Retrospective—Cohort	USA	Myriad Genetic Laboratories database	3828	1397	56 (IQR 48–64)	Caucasian 59.3%	Increased risk of any UTC (RR 4.1 vs. 1.2%, *p* < 0.0001; OR 3.22 (2.66, 3.99), *p* < 0.0001) Increased risk of bladder cancer (RR 1.8 vs. 0.4, *p* < 0.0001) and kidney cancer (RR 1.3 vs. 0.7%, *p* < 0.0003)
							African American 4.8%	
							Asian 2.9%	
							Other 13.8%	
							Missing/no response 19.1%	

Legend: 95% CI = 95% confidence interval; F = female; GU= genitourinary; HNPCC = hereditary nonpolyposis colon cancer; IQR = interquartile range; M = male; MMR = mismatch repair; OR = odds ratio; RCC = renal cell carcinoma; RR = relative risk; SD = standard deviation; SIR = standardised incidence ratio; UTC = urinary tract cancer.

**Table 3 cancers-15-00506-t003:** Risk of bias analysis using the Cochrane risk of bias tool for cohort studies.

	D1	D2	D3	D4	D5	D6	D7	D8
Aarnio et al. (1999) [22]	A	A	B	C	B	B	B	N/A
Pande et al. (2012) [23]	B	B	B	C	B	B	B	N/A
Engel et al. (2012) [24]	A	A	B	C	B	B	B	N/A
Win et al. (2012) [25]	D	B	B	C	B	C	B	N/A
Wischhusen et al. (2020) [4]	A	A	B	C	B	B	B	N/A

Legend: A—Definitely yes (low risk of bias); B—probably yes; C—probably no; D—definitely no (high risk of bias). D1—Was the selection of exposed and non-exposed cohorts drawn from the same population? D2—Can we be confident in the assessment of exposure? D3—Can we be confident that the outcome was not present at start of study? D4—Did the study match exposed and unexposed for all variables that are associated with the outcome of interest or did the statistical analysis adjust for these prognostic variables? D5—Can we be confident in the assessment of the presence and absence of prognostic factors? D6—Can we be confident in the assessment of outcome? D7—Was the follow-up of cohorts adequate? D8—Were co-interventions similar between groups?

## Data Availability

Template data collection forms, data extracted from included studies, and data used for analyses can all be made available upon request.

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
