# Peer review of "Relative Risk of Bladder and Kidney Cancer in Lynch Syndrome: Systematic Review and Meta-Analysis"

_cancers, 2023, doi:10.3390/cancers15020506_

Round 1

Reviewer 1 Report

The article is probably the first serious systematic review and meta-analysis dealing with the relative risk of developing bladder and kidney cancer in adults with confirmed Lynch Syndrome germline mutations. Results are very interesting and methodology is firm. Concgratulations.

I have a few minor concerns.

The first one has already been addressed by the authors and is the uncertainty of the histology of the renal mass reported in the studies evaluated. In fact, renal cancer is different to kidney cancer because several very different histologies compose kidney neoplasm. The authors should mention this drawnback and should also analyse (if possible) the results reported on renal cell carcinoma (renal cell clear cell carcinoma, RCCCC) in patients with Lynch syndrome. It should be also stated that renal cancer is a very heterogeneous disease, and this intrinsic limitation could hamper the results. To make matters worse many registries do not adequeately separate CCRCC and urothelial carcinoma of the kidney. THis should also be registered at least in the limitations of the study.

Another question is why was prostate cancer excluded from the analysis?

Finally, I would like to see somewhere in the manuscript that, appart from Lynch syndrome germline mutations in MM-repair genes, the disese is sometimes cause by epigenetic silencing. I would like to know if there is a different pattern of neoplastic disease outside the upper urinary tract for one or the other pathways of the disease.

Another observation is that the authors should try to make their tables and figures more consistent with the MDPI template.

Reviewer 2 Report

Well written paper, minor comments:

P4 L65 definition of „sentinel urinary malignancy”?

P6 L161-163 “However, in a study of 321 LS carriers, they found that 161 of the 14/321 patients with bladder cancer, were non-invasive (pTa/T1) in 93% and high- 162 grade bladder in 71% of this cohort with a median follow-up period of 9-years [11].” The sentence is difficult to understand, please reconsider.

P7 L202 ff please cite and briefly discuss this recent work on MSI-screening: https://doi.org/10.1016/j.urology.2022.02.006 https://doi.org/10.1016/j.csbj.2021.08.037 https://doi.org/10.1016/j.csbj.2021.08.037

P7 L206-208 “Only 2/17 studies from a pooled dataset reported a sensitivity of 63% and specificity of 85 and 88% for PCR and IHC, respectively in LS-UTUC [37,38,46].” This sentence seems to indicate, that IHC has the same sensitivity and higher specifity than PCR. If this is the case, please comment on why PCR-testing is recommended before IHC (figure 3).

P11 L302-303 don´t confuse MMR-testing (IHC) and genetic testing for MSI (PCR)
